# ADJUSTING PRETRAINED BACKBONES FOR PERFORMATIVITY

## ABSTRACT

With the widespread deployment of deep learning models, they influence their environment in various ways. The induced distribution shifts can lead to unexpected performance degradation in deployed models. Existing methods to anticipate performativity typically incorporate information about the deployed model into the feature vector when predicting future outcomes. While enjoying appealing theoretical properties, modifying the input dimension of the prediction task is often not practical. To address this, we propose a novel technique to adjust pretrained backbones for performativity in a modular way, achieving better sample efficiency and enabling the reuse of existing deep learning assets. Focusing on performative label shift, the key idea is to train a shallow adapter module to perform a *Bayes-optimal* label shift correction to the backbone's logits given a sufficient statistic of the model to be deployed. As such, our framework decouples the construction of input-specific feature embeddings from the mechanism governing performativity. Motivated by dynamic benchmarking as a use-case, we evaluate our approach under adversarial sampling, for vision and language tasks. We show how it leads to smaller loss along the retraining trajectory and enables us to effectively select among candidate models to anticipate performance degradations. More broadly, our work provides a first baseline for addressing performativity in deep learning.

## 1 INTRODUCTION

Machine learning models have been experiencing a growing adoption for automated decision-making. High-stake applications necessitate models to generalize beyond the training distribution and perform robustly over distribution shifts. A prevalent but often neglected cause of distribution shift is the model deployment itself. When informing down-stream decisions, and actions, the predictions of machine learning models can change future data. Such patterns are ubiquitous in social settings, where algorithmic predictions impact individual expectations, steer consumer choices, or inform policy decisions. Similarly, standard community practices can lead to future data depending on the deployment of past models; this can be through data feedback-loops (Taori & Hashimoto, 2023), active learning pipelines (Settles, 2009), and dynamic benchmarks (Nie et al., 2020). Performative prediction (Perdomo et al., 2020) articulates how this causal link between predictions and future data surfaces as distribution shift in machine learning pipelines.

It is inevitable that repeatedly ad-hoc trained models become suboptimal after deployment under performativity (Kumar et al., 2022; Recht et al., 2019; Taori et al., 2020). Thus, a natural question to ask is—can we learn to foresee these shifts? Of course, in full generality performative shifts can be arbitrarily complex. However, given a low-dimensional sufficient statistic for the shift (e.g. class-wise accuracies, group accuracies etc.), Mendler-Dünner et al. (2022) show that a data-driven approach can be successful in anticipating performativity. Once the relevant mechanism mapping the model statistic to the induced data is learnt, shifts can be anticipated and the performative prediction problem can be solved offline (Kim & Perdomo, 2023). While this approach is appealing theoretically, a demonstration of the practical feasibility in the regime of deep learning was still missing.

In particular, existing approaches (e.g., Mendler-Dünner et al., 2022; Kim & Perdomo, 2023) learn predictive models from scratch, assuming access to performativity-augmented datasets that contain statistics about the deployed model, in addition to feature-label pairs. In large-scale deep learning, this approach to anticipating performativity has two fundamental practical limitations. First, large-

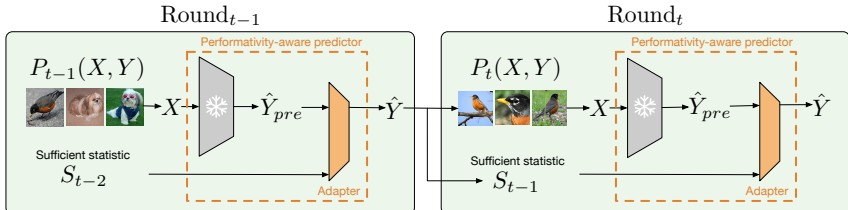

Figure 1: Setup: In each round a model is deployed to make predictions $\hat{Y}$ over $P_t$. These predictions give rise to a new distribution $P_{t+1}$. To achieve high accuracy after deployment, we equip existing backbones with an adapter module to build a performativity-aware predictor. The adapter module seeks to predict the next distribution based on the sufficient statistic $S$ for the shift, and adjusts the predictions accordingly. Under performativity the $S$ is a function of the deployed model.

scale models are extremely data-hungry when trained from scratch, and performativity-augmented datasets are hard to gather and they are not yet widely available. Second, existing pre-trained models only process raw features and they are not compatible with this paradigm, preventing the utilization of valuable data resources and existing open source models. In this work we provide the first practical approach to building performativity-aware deep learning models around pre-trained backbones.

## 1.1 OUR WORK

We propose a pipeline to adjust deep learning predictions for performative label shift. This refers to a setting where the deployment of a model changes the class proportions in future rounds. This setting is particularly relevant for vision and language tasks, where pretrained backbones prevail. For instance, adaptive data collection (Shirali et al., 2023; Nie et al., 2020) and performance-dependent participation (Liu et al., 2018; Ensign et al., 2018) are instances of this problem which has been under ongoing investigation (Lipton et al., 2018; Liu et al., 2021a; Garg et al., 2020b).

To anticipate performative label shift, we propose a modular framework to equip existing pre-trained models with a learnable adaptation module. The adaptation module takes the sufficient statistic for the shift, and the pretrained model's intermediate representations as input and outputs adjusted predictions. In the concrete instantiation of performative label shift, the adaptation module learns to predict the label marginals and corrects for performativity post-hoc with a Bayes-optimal correction to the model's logits. More generally, our framework decouples the task of modeling the underlying concept from modeling performative effects. This has the crucial advantage that existing pre-trained models can be used for the former, and only the parameters of the latter are learned from performativity-augmented data, making it more practical and data-efficient.

To evaluate our approach, we draw upon connections between performativity and dynamic benchmarks (Nie et al., 2020) and simulate performative shifts over vision and language tasks through adversarial sampling. Our main empirical findings can be summarized as follows:

- We demonstrate that our proposed adaptation module can learn the performative mechanism effectively from **a few performativity-augmented datasets** collected along a natural retraining trajectory.

- The module enables us to adapt the predictor to future data *before* deployment, **significantly reducing performance degradation** due to performative distribution shifts, compared to state-of-the art fine-tuning techniques.

- The readily trained adjustment module is **flexible** in that it can be combined with various pre-trained backbones allowing zero-shot transfer during model updates, e.g., when more performant backbones become available.

- We show that our trained adaptation module can anticipate a model's brittleness to performative shifts before deployment, enabling **more informed model selection**.

In a nutshell, we offer the first baseline to effectively adapt state-of-the art deep learning models to performative distribution shifts. Along the way we highlight connections between performative prediction, state-of-the art fine-tuning techniques and their application in dynamic benchmarking, as well as several interesting opportunities for future work.

## 2 BACKGROUND AND RELATED WORK

Perdomo et al. (2020) introduce the framework of **performative prediction** to study performativity in machine learning. We refer to Hardt & Mendler-Dünner (2023) for a comprehensive overview on related literature. The key conceptual component of the framework is to allow the data distribution to depend on the predictive model. A natural approach to deal with distribution shifts of all kind is to perform naive retraining. Interestingly, such heuristics can converge to equilibria under performativity (Perdomo et al., 2020; Mendler-Dünner et al., 2022; Li et al., 2022; Drusvyatskiy & Xiao, 2023). However, it is known that retraining can lead to suboptimal solutions even after convergence (Perdomo et al., 2020; Miller et al., 2021). Thus, a more ambitious goal is to anticipate performative shifts, instead of solely responding to them (Miller et al., 2021; Jagadeesan et al., 2022). In particular, Mendler-Dünner et al. (2022) suggest treating predictions as features in a machine learning model, assuming that performativity is mediated by predictions. Kim & Perdomo (2023) formalize requirements under which such a model allows for optimizing any downstream loss under performativity, also referred to as an omnipredictor (Gopalan et al., 2022). Both of these approaches require a dataset containing information about the deployed model large enough to train a performativity-aware predictor from scratch. Unfortunately, such data is rarely available in practice, and the paradigm prevents the use of existing pre-trained models and benchmark datasets as they lack such information. To the best of our knowledge, we are the first to offer a solution that allows to build on existing pretrained-backbones towards this goal.

We primarily focus on **label shift** in this work. Label shift refers to the shift of the marginal distribution $P(Y)$, while the class conditionals $P(X|Y)$ remain fixed (Manski & Lerman, 1977; Storkey et al., 2008; Zhang et al., 2013; Lipton et al., 2018). In contrast to prior work on model-induced shifts, focusing predominantly on covariate shift (e.g., Hardt et al., 2016) and concept shift (Mendler-Dünner et al., 2022; Kim & Perdomo, 2023), our focus on deep learning applications puts forth this novel and important dimension of label shift. While concept shift would mean, e.g., a change in the image labeling function, label shift means a change in the sampling procedure, making it much more ubiquitous. Performance degradation due to label shift has been a long-standing problem for computer vision tasks (Menon et al., 2021; Kang et al., 2020; Liu et al., 2019; Cui et al., 2019; Shi et al., 2023; Gao et al., 2024; Zhang et al., 2023), with a plethora of principled approaches to correcting the label shift through unlabeled test data (Alexandari et al., 2020; Garg et al., 2023; 2020a). However, existing efforts do not consider the dynamic interplay between model deployments and induced shifts. Our work aims to address this issue and provide initial empirical baselines.

A **practical setting** where performative label shift surfaces are adaptive data collection settings. Here performativity is a response to the predictive performance of the model. For example in active learning Settles (2009) data samples are collected to obtain information in high uncertainty regimes of the current model. Dynamic benchmarks Nie et al. (2020) suggest designing datasets adaptively to challenge prior models. Approaches to mitigating fairness issues Abernethy et al. (2022); Globus-Harris et al. (2022) suggest collecting data for groups on which the model performs poorly. In all these settings, the shifts are mediated by model performance affecting future data collection. In contrast to tabular data, performative concept shift is less common in image and language settings, whereas covariate and label shift prevail.

Finally, there are various **techniques to address distribution shifts** in deep learning, independent of their origin. Prominent example include full fine-tuning (Kornblith et al., 2019; Kolesnikov et al., 2020; Zhai et al., 2020), partial adaptation (Chen et al., 2020; Houlsby et al., 2019), last-layer retraining (Kirichenko et al., 2023; Rosenfeld et al., 2022; Hu et al., 2022; Dettmers et al., 2023), prefix-tuning Liu et al. (2022); Jia et al. (2022), unsupervised domain adaptation (Zhang et al., 2013; Ganin et al., 2016; Shu et al., 2018; Sun & Saenko, 2016; Courty et al., 2017) and test-time adaptation (Sun et al., 2020; Wang et al., 2021; Liang et al., 2020; Liu et al., 2021b). Orthogonal to these, continual learning focuses on mitigating catastrophic forgetting (Kirkpatrick et al., 2017) (i.e., knowledge accumulation) while dealing with a stream of data distributions (Castro et al., 2018; Aljundi et al., 2018; Hou et al., 2019; Belouadah & Popescu, 2019; Wang et al., 2024; Mi et al., 2020). None of these methods is designed to address shifts proactively. They all need to observe the induced distribution before adaptation and, thus, inevitably suffer from performance degradation due to performative shifts. By training the model to learn how to perform an adaptation before a performative shift occurs, our work takes a first step into a widely unexplored new direction to improve predictive performance under distribution shifts of known cause.

## 3 ANTICIPATING PERFORMATIVITY

Performative distribution shifts are caused by model deployment. Thus, having access to the right statistic about the model is in principle, sufficient to foresee performative shifts. This is the core idea making it possible to anticipate performativity, in contrast to arbitrary distribution shifts. Making this more practical is the challenge we tackle in this work. Figure 1 illustrates our proposal.

**Problem setup.** We consider discrete time steps, indicating the deployment of model updates. In each step $t \geq 0$, first, a dataset of feature label pairs $(X, Y)$ is collected. We consider a classification setting with $X \in \mathbb{R}^d$ and $Y$ taking on $K$ discrete values. We use $P_t$ to denote the distribution over data points at time step $t$. Then, a new model $f_t$ is trained to predict $Y$ from $X$. The model $f_t$ is deployed and $t$ is incremented. The new distribution $P_{t+1}$ is fully characterized by a sufficient statistic $S_t$, which is a function of $f_t$ and $P_t$. This corresponds to a stateful extension of the framework by Perdomo et al. (2020), using a Markovian assumption similar to Brown et al. (2022):

$$P_{t+1}(X, Y) = P(X, Y | S = S_t) \quad \text{with} \quad S_t = \text{Stat}(f_t, P_t) \tag{1}$$

An example of a sufficient statistic could be the model predictions over the previous data (Mendler-Dünner et al., 2022; Kim & Perdomo, 2023), or model accuracy across subgroups (Nie et al., 2020). Such statistics are typically significantly lower-dimensional than the raw parameters of $P_t$ and $f_t$ (and avoid explicit parametric assumptions for $P_t$). In the following, we assume that, through expert and domain knowledge, we can specify such a statistic. This means we assume the model developer knows, for example, that predictions are causing the shift, rather than the specifics of the model parameters themselves. We leave for future work the possibility of identifying such statistics from data in settings where such knowledge can not be assumed. Following the notion of independent causal mechanisms (Schölkopf et al., 2021; Peters et al., 2017), we assume that the mechanism underlying the distribution shift is fixed and shifts only manifest through instantiations of $S$.

**Practical challenges.** Given a statistic $S$, anticipating performativity means to predict $Y$ from $X$ taking the instantiation of $S$ into account. This corresponds to learning a performativity-aware predictor of the form

$$f_{\text{perf}} : (X, S) \to Y.$$

Toward this goal, we highlight two important practical challenges:

> **Challenge 1: (Scarcity of performativity-augmented data).** Curating a training dataset of $(X, S, Y)$ pairs for learning $f_{\text{perf}}$ can be prohibitively expensive, as it necessitates exposing the environment to models associated with different statistics $S$ and pooling the obtained data together for training. The complexity of gathering such datasets is insurmountable for high-dimensional data such as images and text and drastically increases training costs.

> **Challenge 2: (Compatibility with existing backbones).** The function $f_{\text{perf}}$ processes performativity-augmented data points $(X, S)$ as its input. This forbids the direct application of existing pre-trained deep learning models to learn $f_{\text{perf}}$, as they typically do not include a feature about the statistic $S$ related to the dataset collection as their inputs.

### 3.1 A MODULAR ADAPTATION ARCHITECTURE

Our goal is to develop an architecture to model $f_{\text{perf}}$ that uses $f_{\text{pre}}$ as a building block. That is, we consider functions of the form

$$f_{\text{perf}}(X, S) = F(\{f_{\text{pre}}^{(k)}(X)\}_{k \geq 0}, S), \tag{2}$$

where $f_{\text{pre}}^{(k)}$ denotes the pre-trained model's representation at layer $k$. The adapter module $F$ acts on top of the pre-trained backbone $f_{\text{pre}}$, potentially accesses its intermediate layer representations, and incorporates the statistic $S$ to adjust the model's outputs for performativity.

The adapter module reduces to a scalar function if it operates only on top of the pretrained model's predictions, such as the case for self-negating and self-fulfilling prophecies (Bezuijen et al., 2009; Arrow, 2012), or reflection effects (Manski, 1993). At the same time, the mechanism mapping $X$ to $Y$ could be arbitrarily complex, and $X$ be high dimensional, such as for image or text. Thus, decoupling the feature extraction step from the performative mechanism can come with a significant

reduction in complexity for learning the latter, using $f_{\mathrm{pre}}$ as a building block, instead of learning both jointly. Typically, the adapter is given access to more layers of the backbone for the sake of expressivity. At the extreme it get access to the backbone's input, allowing it to learning the performativity-aware predictor from scratch. With access to more information, the complexity, as well as data requirements for learning the adapter module will naturally increase, offering a useful lever to strategically trade-off assumptions and evidence, and to adapt the module to the availability of performativity-augmented data.

### 3.2 Anticipating performative label shifts

Label shift focuses on the effect of deploying $f_t$ on the marginal distribution $P_{t+1}(Y|S_t)$ (Zhang et al., 2013; Lipton et al., 2018). For a discrete classification task, the marginal over the outcome can be concisely represented with a probability vector $\Lambda \in \mathbb{R}^K$ where $K$ denotes the number of classes, and each entry of $\Lambda$ specifies the corresponding class probability. Thus, anticipating performativity is equivalent to anticipating changes to $\Lambda$.

At the core of the adapter module is a neural network $T : S \mapsto \hat{\Lambda}$ that predicts the label marginals $\Lambda$ from the sufficient statistic $S$. These estimates can be used to anticipate the deployment of future models and adapt the predictions by accessing the pretrained-model's logits. More formally, we implement the following adjustment:

$$f_{\mathrm{perf}}(X, S_t; T) = \arg\max_i \lambda_i(S) \cdot [f_{\mathrm{pre}}(X)]_i \quad \text{with} \quad \lambda_i(S) := \frac{[T(S)]_i}{\Lambda_i^{\mathrm{pre}}}, \tag{3}$$

where $\Lambda^{\mathrm{pre}}$ is denotes the label marginals over the training data of $f_{\mathrm{pre}}$. This expression fully decouples the mechanism underlying the shift from the feature-extraction part on the input. The next result shows that for a well trained $T$ and a good pretrained model, such an adjustment can indeed be optimal under label shift.

**Proposition 3.1.** *Assume the pretrained model $f$ accurately represents the likelihood of the training data. Then, if performativity only surfaces in the marginal $P(Y)$, and $P(Y|X)$ is unaffected by performativity, there exists a predictor $T$ such that $F$ recovers $f_{\mathrm{perf}}$.*

*Proof.* Let $T(S) = P(Y|S)$ and $f(X) \propto P_{\mathrm{pre}}(Y|X)$. Then, following (Saerens et al., 2002; Royer & Lampert, 2015; Menon et al., 2021) we have

$$\lambda_i(S)[f(X)]_i \approx \frac{P_t(Y=i)}{P_{\mathrm{pre}}(Y=i)} \cdot P_{\mathrm{pre}}(Y=i|X) = \frac{P_t(X)}{P_{\mathrm{pre}}(X)} \cdot P_t(Y=i|X) \propto P_t(Y=i|X). \tag{4}$$

and hence the adjustment in (3) is Bayes-optimal under label shifts. □

**Dynamic benchmarking.** Our running example for performative label shift is the use case of dynamic benchmarks (Shirali et al., 2023; Nie et al., 2020). Dynamic benchmarks are a recent and popular way to assess and compare the performance of predictive models across multiple phases, where data collection is performed repeatedly with respect to the model performance. The aim is to challenge the model to be better at places where its performance is lacking Kiela et al. (2021). Model updates and the data collection phases follow each other, creating a feedback loop between model performance and data distribution through adversarial sampling.

**Self-selection.** An alternative mechanism leading to opposite dynamics could be caused by model's poor performance on certain classes or subgroups. These negatively impacted users disengage from the data ecosystem, causing representational disparities in the data (Horowitz et al., 2024; Koren, 2024), which can further amplify through retraining (Hashimoto et al., 2018). Both examples are natural use-cases of performative labels shift, where the next round's label proportions are impacted by the model's performance in the current round.

### 3.3 Learning adapter module along the retraining trajectory

Algorithm 1 illustrates a muti-step protocol for training the neural network $T$ to predict the next round's label marginals. In each round fresh data under the deployment of a new model is collected and used to update $T$. More specifically, in each round, we collect the statistic $S_{t-1}$ of the deployed model $f_{t-1}$, together with the induced label marginals $\Lambda_t$ over $P_t$ and store it in a memory buffer to

**Algorithm 1: Building a performativity-aware predictor.**

**Input** : Frozen pre-trained model $f_{\text{pre}}$ and training label marginals $\Lambda^{\text{pre}}$. Randomly initialized adapter $T_0$, empty memory buffer $\mathcal{M}$. Initial distribution $P_0$

1 Deploy $f_0 = f_{\text{pre}}$
2 $S_0 \leftarrow \text{Stat}(f_0, P_0)$
3 **for** *round $t$ in $1, 2, 3, \ldots, T$* **do**
4     Observe samples from $P_t$

5     Update adaptor module:
6     $\Lambda^{(t)} \leftarrow$ marginals evaluated on observed samples
7     Write $(S_{t-1}, \Lambda^{(t)})$ to $\mathcal{M}$
8     $T_t \leftarrow$ update $T_{t-1}$ using gradient descent doing a pass over $\mathcal{M}$

9     Anticipate model deployment:
10     Let $\tilde{S}_t$ be the sufficient statistic to anticipate
11     $f_t \leftarrow$ construct a Performativity-aware Predictor from $T_t(\tilde{S}_t)$ as in (3)
12     $S_t \leftarrow \text{Stat}(f_t, P_t)$
13     deploy $f_t$

**Output:** Performativity-aware Predictor $f_{\text{perf}} = f_T$;

learn the predictor $T$ in a supervised manner. Algorithm 1 aggregates data along a natural retraining trajectory, where the previous round's adjusted predictor is deployed repeatedly. This is reflected by $\tilde{S} = \text{Stat}(f_{t-1}, P_{t-1})$ defining the next distribution.

**End product.** Our algorithm outputs the trained module $T$ that serves to construct a performativity-aware predictor and to anticipate the performative label shift of future deployments. Once $T$ is known, the consequences of a model deployment can be anticipated before actually putting it out in the wild, simply by feeding the model's statistic into the adjustment module to predict the consequences. While we focus on predictive accuracy as a metric in this work, the same procedure could be used to directly measure class imbalances after deployment, and account for the desire to reflect different groups equally well in the data (Wyllie et al., 2024), or other societal desiderata (Davis et al., 2021; Barocas et al., 2023).

## 4 EXPERIMENTS

We empirically investigate the performance of our adapter module under performative label shift for vision and language classification tasks. For vision, we evaluate our model on CIFAR100 (Krizhevsky et al., 2009), ImageNet100 (Chun-Hsiao Yeh, 2022), and TerraIncognita (Beery et al., 2018). For language, we use Amazon (Ni et al., 2019) and AGNews (Zhang et al., 2015). We evaluate the performances of different baselines in a semi-synthetic setting where we simulate model deployments and performative shifts across multiple rounds of retraining.

**Baselines.** We use three different baselines for adjusting a model to performative distribution shifts: *Oracle Fine-tuning*, *Oracle Distribution*, and *No Adaptation*. All of them start with the deployment of a pretrained model and then tackle performative shifts in their own way.

- *Oracle Fine-tuning* adapts the pretrained model by training it with complete information about current round's $(x, y)$ pairs for 25 epochs after observing the shift. While this approach ensures convergence on the available data and allows the model to continuously learn from an expanding number of samples across rounds, it may be computationally costly and potentially overfits to the current distribution, which increases its sensitivity to distribution shifts. In other words, *Oracle Fine-tuning* updates $f_t$ in each step to fit the current distribution $P_t(X, Y)$.

- *Oracle Distribution* uses the true label marginals to adjust the pretrained model's predictions instead of the estimates from the adapter module. It serves as an upper bound.

- *No Adaptation* uses a fixed pretrained model without making any adjustments for the performative distribution shifts over rounds. This baseline provides a reference point for evaluating the value of adaptation strategies in handling performative shifts.

Table 1: *Anticipating performative label shift*. The table reports the performance of different models on CIFAR100. Performance is measured on a balanced dataset (pre deployment), after the shift caused by the model (post deployment), and compared with the performance estimate of the *PaP* module. Our module, correctly anticipates the model ranking which would be incorrect if model selection was performed with the accuracy according to pre deployment, ignoring performativity.

| Model | Pre deployment | Post deployment | PaP estimate |
|-------|----------------|-----------------|--------------|
| Model 1 | 82.60 ①  | 72.50 ③ | 76.42 ③ |
| Model 2 | 82.12 ② | 78.72 ① | 77.66 ① |
| Model 3 | 81.80 ③ | 75.90 ② | 77.53 ② |

## 4.1 PERFORMATIVE LABEL SHIFT

We simulate performative label shift caused by a model's predictive accuracy in previous rounds, as observed in the context of dynamic benchmarks (Shirali et al., 2023), adversarial sampling (Nie et al., 2020) and self-selection (Horowitz et al., 2024), see discussion in Section 3.2. Thus, we use the model's class-wise accuracy as a sufficient statistic for the shift, i.e.,

$$S_t := [\text{Acc}_t[0], \text{Acc}_t[1], ..., \text{Acc}_t[K-1]], \tag{5}$$

where $\text{Acc}_t[i]$ represents the accuracy of class $i$ after model deployment at time step $t$.

To simulate the performative effect, we pass the current rounds class accuracy through a Softmax to obtain the proportion of each class in the next round. Specifically, for any class $i$ the class proportion in round $t + 1$ is chosen as

$$P_{t+1}(Y = i|S) = \exp(S_i/\tau)\Big[\sum_{j=1}^{K} \exp(S_j/\tau)\Big]^{-1}, \tag{6}$$

where $\tau \neq 0$ parameterizes the shift and $\text{Acc}[i]$ represents the accuracy of class $i$. The equation in (6) is a modeling choice for simulating the effect which is hidden to the algorithm. For $\tau < 0$ it emulates the adversarial setting where classes that achieve high accuracy in the past round will diminish in the next round, and vice versa. In contrast, for $\tau > 0$ classes with higher accuracy would be stronger represented in the next round, accumulating mass in small regions of the input space, making the task trivial.

**Strength of performativity.** We use the parameter $\tau$ to simulate different strengths of performativity. We selected three different values for $\tau$, chosen to induce absolute accuracy drops of $2\%$, $5\%$, and $10\%$ after observing a balanced distribution for each model and freeze $\tau$ thereof. We refer to these as the low, moderate, and high shift scenarios, respectively.

**Evaluation metric.** We simulate each model's trajectory (i.e., the repeated training and evaluation over multiple rounds/steps) for 200 steps and we repeatedly evaluate the accuracy after deployment, denoted as $\text{Acc}_t = \text{Acc}(f_t, P_{t+1})$. Note that this implies that all models encounter different distributions after the first round. For reference we evaluate all models on the initial balanced distribution at $t = 0$. For each model, we compare the performance on the trajectory of distributions induced by the respective model to avoid bias toward any particular approach. In addition, we also analyze the utility of the reusable adapter module resulting from the *PaP* procedure.

## 4.2 EMPIRICAL FINDINGS

We conduct experiments with performative label shift, as instantiated above, with varying parameters, applied to data of different modalities.

**Retraining trajectory.** Figure 2 shows the experiments conducted on ImageNet100 and CIFAR100 datasets. We observe that our adapter module (*PaP*) demonstrates comparable performance to *Oracle Fine-tuning* even in the low shift scenario, yet without the more resource-intensive demands in terms of time and compute. High shift scenarios reveal the sensitivity of fine-tuning strategy even when it has complete information about the samples collected throughout rounds. Adopting fine-tuning makes the model lean heavily towards the previous round's class marginals, making the

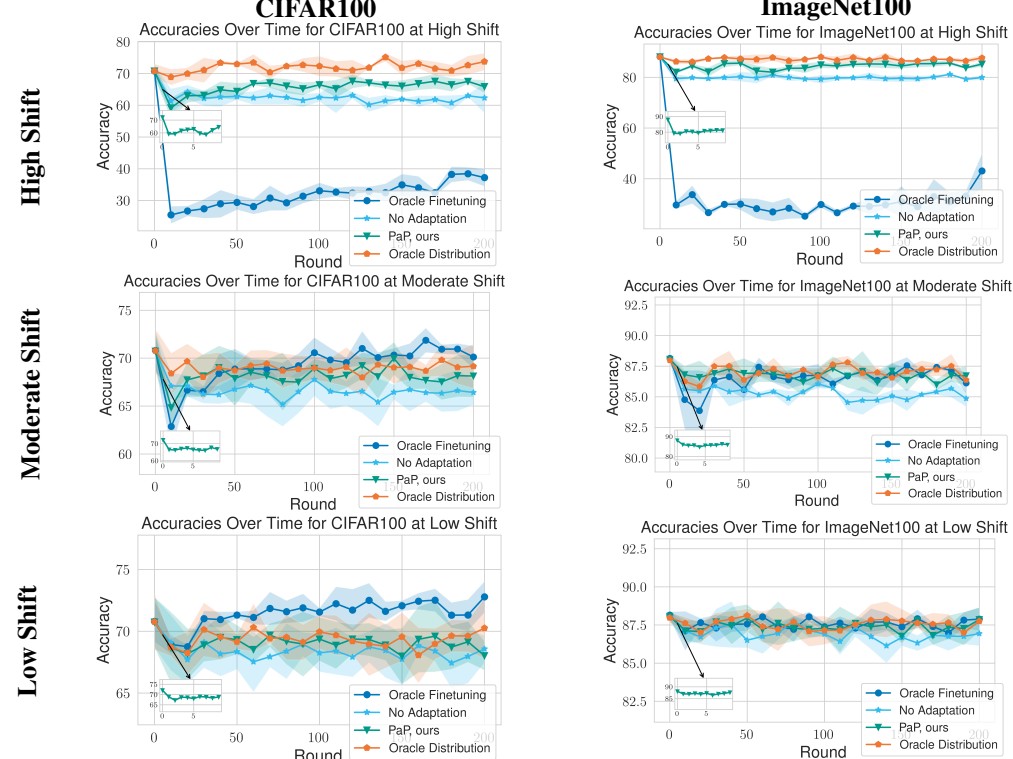

Figure 2: *Accuracy along retraining trajectory for vision tasks.* Each method starts from the same pretrained model, evaluated on the balanced dataset at $t = 0$. Starting from $t = 1$, we simulate 200 rounds of deployments with performative shift of varying strength. The *Performative-aware Predictor (PaP)* performs well even under the high shift scenario, approaching Bayes-optimal update performance as it is trained over rounds. The inset plot zooms in on the performance up to the first checkpoint. As it learns the structure, it typically adapts to the shift within the first 10 updates.

model vulnerable to the upcoming distribution shift. Instead, *PaP* leverages the causal relationship between performance and subsequent distribution, relying solely on the label marginals from previous rounds to model this relationship. In Figure 5 we show the average accuracy improvement of different approaches over *No Adaptation*. We see significant average accuracy improvements of 3.31% and 4.25% for *PaP* on CIFAR100 and ImageNet100, respectively. While these improvements fall short of the Bayes-optimal update's enhancements of 9.4% and 6.88% on the same datasets, they underscore the effectiveness of our adapter module in approximating the causal mechanism. Additionally, its ability to achieve such improvements while being computationally and informationally efficient highlights its adaptability across different shift settings.

Similar gains can be observed on language tasks. Figure 3 illustrates the performances of the different baselines on Amazon and AGNews datasets. Again, it can be seen that high shift scenarios hinder the *Oracle Fine-tuning* performance, failing to anticipate the next distribution similar to the vision case. Moreover, the results reveal that our *Performativity-aware Predictor* steadily approaches the performance of the *Oracle Distribution* over time, as the model learns the inherent relationship between class accuracies and subsequent label distributions. We inspect the learning curve of the adapter in Appendix A.2. Looking at the comparison to *No Adaptation* in Figure 5 we see an average accuracy gain of 2.83% and 1.3%.

**Modularity and zero-shot model updates.** We demonstrate the modularity of our approach in Figure 4 under high label shift. We first trained the adapter module combined with a ResNet18 backbone at high shift on ImageNet100. Then, using the same pretrained frozen adapter, we simulated 200 rounds starting with ResNet18. Over the rounds, we switched the deployed backbone from ResNet18 to ResNet34, and then from ResNet34 to ResNet50. Since the predictor captures the inherent relationship between the sufficient statistic (i.e., class level accuracies) and the label

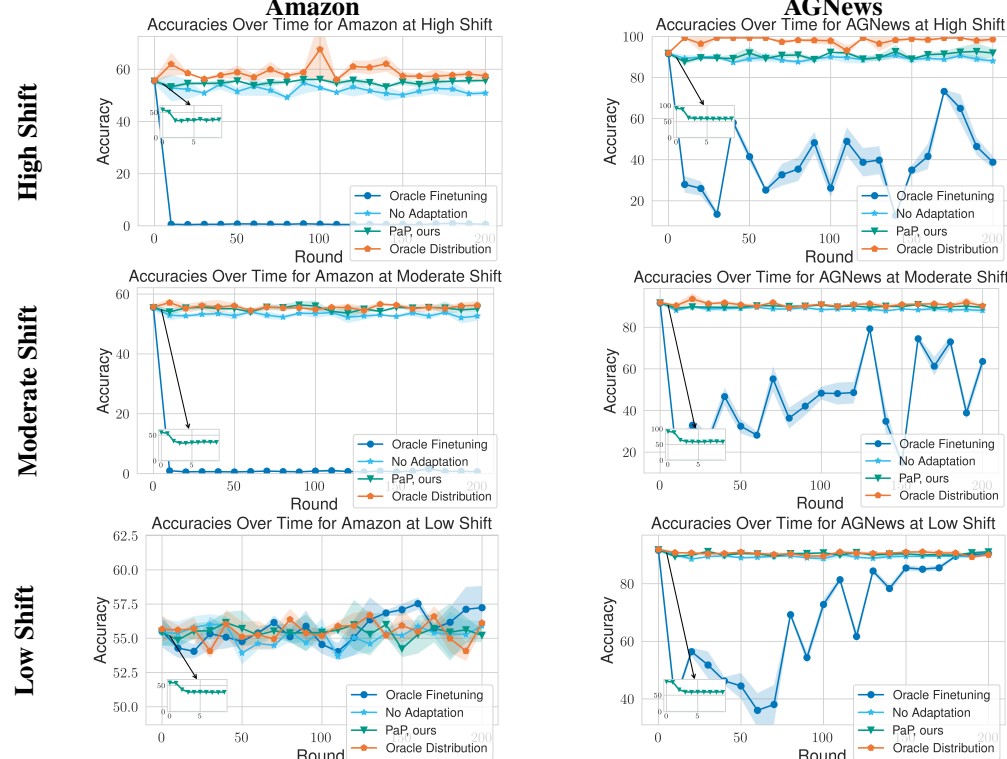

Figure 3: *Accuracy along retraining trajectory for language tasks*. The *Oracle fine-tuning* method is more sensitive to shifts in language datasets. Again, $t = 0$ refers to the balanced training accuracy. Similar to the vision case, the *Performative-aware Predictor (PaP)* performs well under different shift scenarios, increasing its proximity to the Bayes-optimal *Oracle distribution* performance as it is trained over rounds. The inset plot provides a detailed view of the initial performance, focusing on the model's learning curve within the first 10 updates.

marginals, it is not coupled with the specific model it attaches to and continues to improve with its corrections. Consequently, practitioners can update the current model if a more suitable one becomes available at *any time* during deployment cycles, without the need to retrain the predictor.

**Anticipating performativity.** We demonstrate that the learnt adapter module *PaP* can effectively anticipate the future performance of a model before its deployment, providing valuable information for model selection. Our experiment involves training various models with different random initializations. For each model, we evaluate its performance on a balanced dataset (pre deployment), then deploy it and measure performance again on the induced data (post deployment). In parallel, we use our learnt adapter model to predict post deployment performance, given only the sufficient statistic, and sample access to the current distribution. In Table 5, we compare our anticipation with the actual performance. We can observe that our approach provides a much better estimate than the initial performance. Importantly, the ranking based on our estimates exactly matches the true shift performance ranking, which cannot be inferred from first round performances alone. For example, Model 1 initially outperforms Models 2 and 3. However, the nature of the performative shift affects Model 1 more significantly, resulting in worse post-shift performance compared to Models 2 and 3. Using *PaP*, one can infer that Models 2 and 3, despite having worse performance on the current distribution, are more robust against the performative shift.

**Beyond label shift.** As a more general setting, we combine label shift with domain shift. For illustration purposes, we simulated an almost extreme scenario. Specifically, we randomly selected two domains and sampled data points exclusively from these domains. Using the TerraIncognita dataset (Beery et al., 2018) and employing the same experimental setting over 200 rounds, we evaluate the average accuracies across rounds with their standard errors in this scenario. The *Performativity-aware Predictor* achieves an average accuracy of $78.25 \pm 3.24$, outperforming the *No Adaptation*

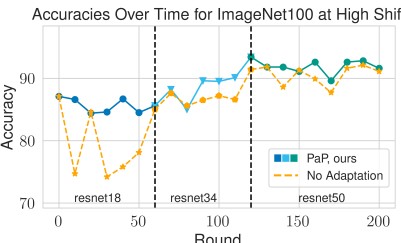

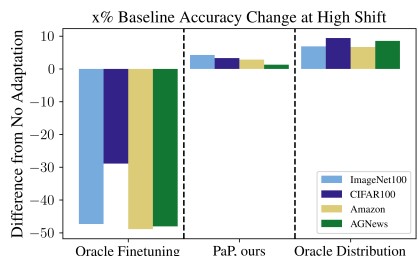

Figure 4: *Modularity of the architecture.* We conduct a model switching experiment where we replace the backbone within PaP. *PaP* still outperforms the *No Adaptation* baseline consistently, even with models it wasn't originally trained with.

Figure 5: *Anticipating performativity.* Average performance gain over *No Adaptation* in high shift scenarios. *Oracle fine-tuning* performs significantly worse than *No Adaptation*, as it does not anticipate the shift. In contrast, *PaP* achieves consistent gains and performs comparable to the oracle baseline.

case with $75.39 \pm 3.32$ and the *No Adaptation (only label shift)* case with $75.89 \pm 1.31$. The high standard error shows that the presence of simulated domain shift results in higher fluctuation in performance, reflecting the extremity of our simulation. However, assessing the average accuracy performance reveals that the effect of additional domain shift on performance is not highly significant. Furthermore, one can see the effectiveness of using the adapter module compared to *No Adaptation* when both types of shifts are present. *PaP* outperforms no adaptation cases by almost 3%, both in the presence of label shift alone and when both shifts are combined. This demonstrates that our adapter module designed to correct for performative label shift remains effective even in the presence of additional sources of shifts on the input distribution.

## 5 CONCLUSION

This work investigates performative prediction in deep learning. We design the first practical algorithm to adjust pre-trained models for performativity that is compatible with existing deep learning assets. We motivate the use of modular architectures to increase data efficiency and evaluate our approach under performative distribution shifts arising in typical dynamic benchmark settings. On multiple vision and language datasets with different types of shifts, we observe consistent performance gains along the retraining trajectory compared to standard baselines for the same adjustment module applied to different backbones. Finally, we illustrate how the adapter can be used for model selection under performativity to enable more informed model deployments and anticipate unwanted consequences.

**Limitations and extensions.** Overall, our work is the first tackling performativity in deep learning. Thus, there are countless possible extensions of our method. We demonstrated the feasibility of designing a modular architecture in the context of performative label shift by accessing the logits of the pre-trained models. An interesting and natural direction could be to capture and adjust representations, closer to the input level. This would allow to account for more complex shifts, and offers a natural lever to trade off expressivity of the adapter and sample requirements. Further, our approach critically assumes known statistics, which are easily encoded in the label shift setting we consider, and easy to reconstruct with minimal knowledge about the paradigm. An interesting extension is to learn such statistics, perhaps leveraging causal representation learning tools (Schölkopf et al., 2021). Besides being more general, it could also serve open vocabulary tasks (Radford et al., 2021), where even labels shifts would be challenging to characterize with a finite dimensional vector, or even generative modeling. Another unanswered question is to derive theoretical guarantees for learning the underlying performative mechanism such as causal identification guarantee, similar to Mendler-Dünner et al. (2022), as well as sample complexities. These results can potentially guide more data-efficient algorithms, or more effective strategies to select the sequence of models to deploy during the training phase of the adapter module, akin to Jagadeesan et al. (2022).

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

# A APPENDIX

## A.1 IMPLEMENTATION DETAILS

Here we report implementation details omitted from the body of the paper due to space limitations. We first give details about the datasets used, then explain training details of our approach.

Table 2: Hyperparameter Configurations for Different Datasets

| Dataset | Backbone | Temperatures | Learning Rate | Batch Size | Epochs & Rounds | Optimizer |
|---------|----------|--------------|---------------|------------|-----------------|-----------|
| CIFAR100 | ResNet18 | 0.1/0.3/0.5 | $1e-3$ | 16 | 25/200 | SGD |
| ImageNet100 | ResNet18 | 0.1/0.3/0.6 | $1e-3$ | 16 | 25/200 | SGD |
| TerraIncognita | ResNet18 | 0.1/0.2/0.8 | $1e-3$ | 16 | 25/200 | SGD |
| Amazon | DistilBERT | 0.05/0.1/0.45 | $1e-5$ | 24 | 3/200 | AdamW |
| AGNews | DistilBERT | 0.01/0.025/0.05 | $1e-5$ | 24 | 3/200 | AdamW |

**Dataset Details**

- **ImageNet100:** The ImageNet100 dataset Chun-Hsiao Yeh (2022) is a subset from the ImageNet Large Scale Visual Recognition Challenge 2012. It contains random 100 classes, each having 1350 samples with resolution $3 \times 224 \times 224$.

- **CIFAR100:** The CIFAR100 Krizhevsky et al. (2009) dataset has $60,000$ images with 100 different classes and resolution $3 \times 24 \times 24$.

- **TerraIncognita:** The TerraIncognita dataset Beery et al. (2018) consists of wild animal photographs with 4 domains based on the location where the images were captured. It contains $24,788$ images with a resolution of $3 \times 224 \times 224$ and 10 classes.

- **Amazon:** The Amazon review dataset Ni et al. (2019) is a text classificaiton dataset containing reviews for products together with the scores from the users. It has $4,002,170$ reviews with 5 classes.

- **AGNews:** The AGNews dataset Zhang et al. (2015) consists of a collection of collection $127,600$ news articles with 4 classes.

**Training Details.** We use a train-test-split with ratios 0.4, 0.3 and 0.3 respectively. Each dataset is treated as a data pool for sampling. To compute the initial performance of the pretrained model and generate the first statistic (class-level accuracies) we sample instances using a Dirichlet distribution. Choice of parameter $\alpha$ for the distribution guides the skewness of the initial distribution for the initial model. We set $\alpha = 100$ to evaluate the initial model on a fairly balanced dataset. For each round, we sample $1,000$ train and validation samples and $2,000$ test samples from the data pools to simulate the round. Each iteration of the loop (rounds) follows: (1) evaluation of the existing model on the current distribution, (2) updating the model using the current distribution, (3) computing the statistics using the updated model on the current distribution. The computed statistics in the final step determine the next distribution and these steps are repeated over 200 rounds. Baselines differ based on their approach to step (2). *Oracle Fine-tuning* uses train and validation set to fit to the current distribution. *No Adaptation* skips that step and *Performativity-aware Predictor* adds previous round statistic, current label marginal pair to its memory buffer and make a pass over it to update the label marginal predictor. This memory buffer simulates epoch-like training for the label marginal predictor. Since it passes over the first pair it has added to the memory buffer many times, we apply a scaling to balance sample importance exponentially with a decay factor 0.995. For the vision experiments we use a cosine annealing learning rate scheduler, while for the language datasets we use linear scheduling. Throughout all classification tasks we used a cross entropy loss as the metric. To train the *Performativity-aware Predictor*, we used Adam optimizer with a learning rate of $1e-4$ and KL-Divergence loss. For the model switching experiments, we switched models at rounds 60 and 120 over the course of 200 rounds of simulation.

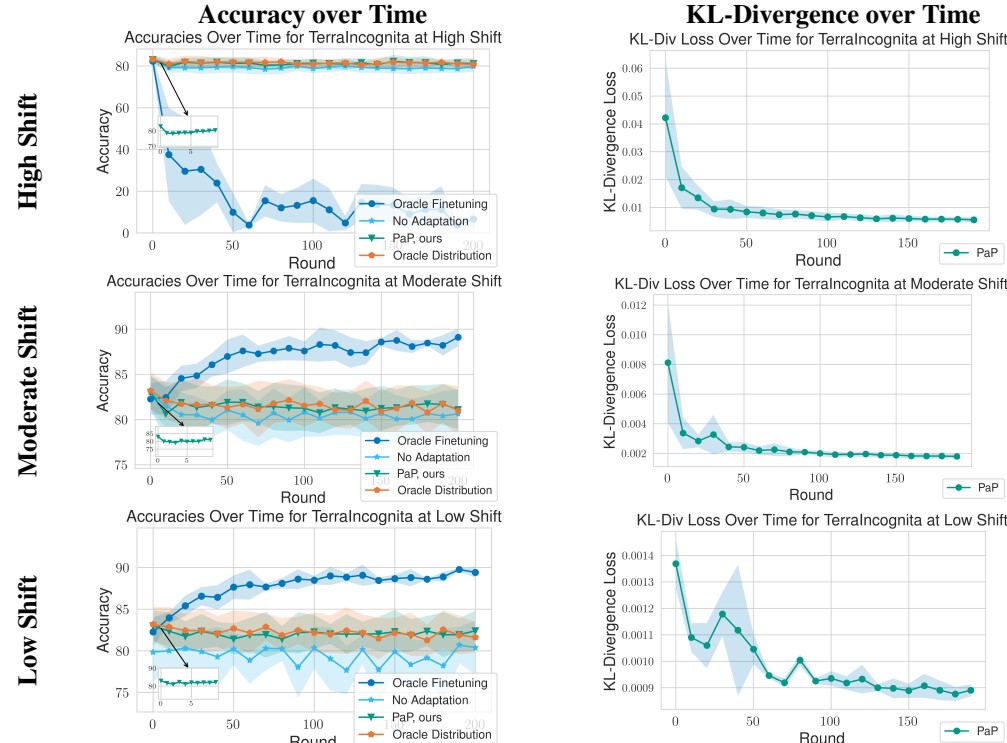

Figure 6: Each row shows the accuracy over time for the TerraIncognita dataset under the corresponding shift, alongside our method's KL-Divergence loss trajectory. The inset plots zoom in to demonstrate our model's adaptation during early rounds. In moderate and low shift scenarios, *Oracle Fine-tuning* scales well with an increased number of training samples, outperforming the *Oracle Distribution*. Consistent with previous experiments, the *Performative-aware Predictor (PaP)* excels under high shift conditions, where *Oracle Fine-tuning* fails. Moreover, across all shift scenarios, *PaP* learns to predict the next distribution's label marginals very accurately after only a few rounds, showcasing its effectiveness.

We use the Transfer Learning Library Jiang et al. (2020) together with PyTorch Paszke et al. (2019) to implement our models. Table 2 shows dataset specific, backbone and optimization-related hyperparameters which are chosen through grid search. All our experiments were run on a local computing cluster using RTX 3090 NVIDIA GPUs with 30 GB of RAM. Although individual jobs are run on a single GPU, we typically used multiple GPUs to run the experiments in parallel.

## A.2 ADDITIONAL EXPERIMENTAL RESULTS

**Label shift on TerraIncognita dataset.** Figure 6 shows a similar pattern as of the previous experiments under the label shift setting in the high shift setting. However, for the moderate and low shift settings, it can be seen *Oracle Fine-tuning* continues to improve its performance over rounds outperforming other baselines. Due to TerraIncognita dataset's fewer number of classes, task for the backbone is easier and it benefits from training more, and there is not enough room for the label shift to confuse the model. Moreover, *Performativity-aware Predictor* learns the prediction of next label marginals accurately after only a few rounds which reflects to its accuracy trajectory. Comparing *Performativity-aware Predictor* with *Oracle distribution* supports that as *PaP* is almost indistinguishable from its upper bound, achieving almost optimal updates.

**Training only the last linear layer does not improve model robustness to performative shift.** Figure 7 demonstrates that training only the classification head is sufficient to incorporate the current distribution's label bias. Although computationally cheaper, this approach suffers from high performative shift similarly to *Oracle Fine-tuning*.

Table 3: Number of trainable parameters and training FLOPs for different models.

| Model | Trainable Parameters | Backward GFLOPs (per round) |
|---|---|---|
| No Adaptation | 0 | 0 |
| PaP | $117,348$ | 0.07 |
| Oracle Fine-tuning (last linear) | $51,300$ | 2.56 |
| Oracle Fine-tuning | $11,740,812$ | $90,804$ |

Table 4: Performance anticipation results on the CIFAR100 dataset. The table reports the performance of models with varying initializations on a balanced set, the next round performance estimate using pretrained *PaP*, and the performance after the true shift.

| Model | First Round | Next Round Estimate | True Shift Performance |
|---|---|---|---|
| Model 1 | 82.60 | 76.42 | 72.50 |
| Model 2 | 82.12 | 77.66 | 78.72 |
| Model 3 | 81.80 | 77.53 | 75.90 |
| Model 4 | 79.56 | 72.95 | 69.40 |
| Model 5 | 78.90 | 72.13 | 66.10 |
| Model 6 | 73.16 | 67.15 | 62.08 |
| Model 7 | 71.74 | 62.75 | 56.58 |
| Model 8 | 71.52 | 62.82 | 64.00 |

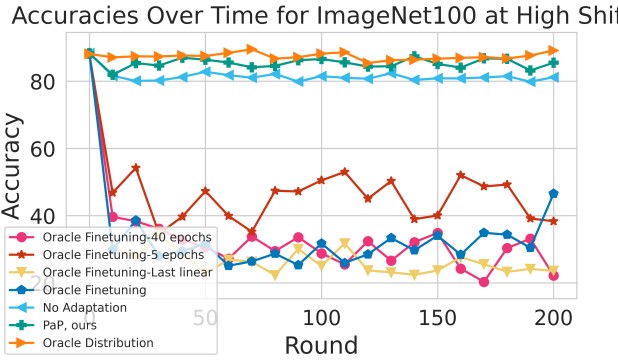

Figure 7: Additional experiments for Oracle-finetuning baseline: including tuning only the last linear layer, training with 5 epochs and training with 40 epochs.

**Varying training epochs on the current distribution for *Oracle Fine-tuning* controls distribution bias incorporation.** Figure 7 illustrates the effect of different training epochs on the current distribution. In our main experiments, we trained until convergence using the current distribution dataset. Setting the number of epochs to 0 reduces *Oracle Fine-tuning* to *No Adaptation*. Our ablation study compares training for 5 and 40 epochs. As expected, in high performative shift scenarios, partial fitting to the current distribution (5 epochs) outperforms full fitting (40 epochs) due to significant differences between consecutive distributions.

***PaP* is a lightweight adaptation module.** Table 3 compares the number of trainable parameters and training FLOPs across baselines. *PaP* offers negligible computational cost while providing (i) adaptation for models under performative label shift, and (ii) evaluation of multiple potential models for deployment selection.

***PaP* can be used for pre-deployment performance evaluation.** Table 4 expands on Table 5, demonstrating that *PaP* rankings closely align with true shift performance. Moreover, PaP provides more accurate estimates of post-shift performance compared to initial model evaluations on the first distribution.

**Different shift mechanism.** Figure 8 shows the performance of different baselines under a different shift mechanism. We used a random neural network with two layers and a ReLU non-linearity

Table 5: *Anticipating performative label shift.* The table reports the performance of different models on ImageNet100. Performance is measured on a balanced dataset (pre deployment), after the shift caused by the model (post deployment), and compared with the performance estimate of the *PaP* module. Our module, correctly anticipates the model ranking which would be incorrect if the model selection was performed with the accuracy according to pre deployment, ignoring performativity.

| Model | Pre deployment | Post deployment | PaP estimate |
|-------|----------------|-----------------|--------------|
| Model 1 | 93.35 (1) | 91.00 (1) | 91.24 (1) |
| Model 2 | 92.50 (2) | 89.3 (3) | 87.28 (3) |
| Model 3 | 92.00 (3) | 89.85 (2) | 88.85 (2) |

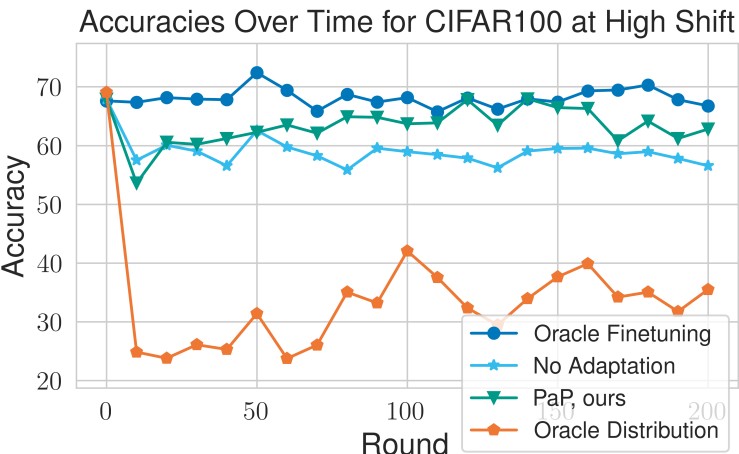

Figure 8: *The figure reports the performance of different baselines on CIFAR100 dataset, under a different shift mechanism. Rather than plain softmax, we created a random 2-layer neural network with a ReLU nonlinearity in between to guide the shift. Here, PaP learns a more challenging non-linear shift function.*

between them to guide the shift. This network takes the sufficient statistic S as input and outputs the class probabilities for the next round. As shown, PaP is able to predict outputs of more complex functions, effectively adapting the backbone's predictions. Our method significantly outperforms the No Adaptation baseline and achieves performance very close to its upper bound. Similar to previous results, Oracle Finetuning struggles with high distribution shifts.

