# OpenReview forum: "Adjusting Pretrained Backbones for Performativity"
_ICLR.cc/2025/Conference — Submitted to ICLR 2025_

### Official Review · Reviewer_Dy2p · 2024-10-31

**Soundness:** 3
**Presentation:** 2
**Contribution:** 2
**Rating:** 5
**Confidence:** 2

**Summary:**

This paper considers a scenario where the input data distribution shifts based on the model's predictions once the trained model is deployed. The authors specifically assume a shift in class marginal distributions and propose a method to learn a shallow adapter module that performs Bayes-optimal label shift correction on the pre-trained classifier. This module adjusts the classifier's logits using a given sufficient statistic. Experiments on CIFAR-100, ImageNet100, Amazon, and AGNews datasets demonstrate that the proposed method can create robust models against tested performative label shifts in both vision and language domains.

**Strengths:**

1. The proposed method is lightweight and easy to implement.
2. The proposed method can create robust models against tested performative label shifts in both vision and language domains.

**Weaknesses:**

1. The assumed performative label shift model in the experiments (Eq. 6) is too simple. It seems possible to model a function that corrects the classifier's logits without using a neural network module. What exactly is the neural network module learning in the proposed method?
2. In real-world scenarios, what situations would the performative label shift tested in the experiments apply to?
3. What is the most common type of performative shift in real-world applications?

**Questions:**

(Copied from Weaknesses)
1. The assumed performative label shift model in the experiments (Eq. 6) is too simple. It seems possible to model a function that corrects the classifier's logits without using a neural network module. What exactly is the neural network module learning in the proposed method?
2. In real-world scenarios, what situations would the performative label shift tested in the experiments apply to?
3. What is the most common type of performative shift in real-world applications?

---

> ### Author Response · Authors · 2024-11-21
> **Response to Reviewer Dy2p**
>
> >**Q1: “... It seems possible to model a function that corrects the classifier's logits without using a neural network module. What exactly is the neural network module learning in the proposed method?”**
>
> In our problem setting we only assumed that there is a relationship between the class-wise accuracies and the next distribution label probabilities (it can be any function that maps class-wise accuracies to label marginals).  We model this cause and effect relationship by parameterizing a neural network.
>
> > **Q2: “In real-world scenarios, what situations would the performative label shift tested in the experiments apply to”**
>
> We assume that the shift is determined by the model performance on individual classes.
> Our main motivation are adversarial sampling strategies. Here, the model performance impacts the decision of a model developer about which datapoints to collect in the next round. The strategy could be adversarial with the goal to challenge models where they perform poorly as in dynamic benchmarks. Alternatively imbalance sampling could be done for the purpose of achieving fairness. It could also happen more passively without the intervention of the developer, but through a self-selection mechanism. Meaning that individuals that experience bad performance drop out of the system due to the limited utility they get from the service. These are all instances of our setup. Typically these mechanisms involve behavioral aspects and they are hard to explain exactly. But our approach can learn from experience some relevant aspects impacting model performance.
>
> > **Q3: “What is the most common type of performative shift in real-world applications?”**
>
> The performative label shift on $ P(Y) $ studied in this work is one of the most common types for applications involving image/text classification, as we discuss in Q4 and the paper.
> For regression tasks on tabular data, perhaps the most common type is the shift on the conditional distribution $ P (Y| X) $. The relevant applications include traffic systems where traffic forecasts influence drivers’ decisions and in turn the future traffic, stock markets where stock forecasts influence the stock values through the holders, and loan interests where individuals predicted to have a high default risk are assigned with a high loan interest and this in turn increases their propensity to default. But in image classification the concept shift  is less natural. Please kindly refer to \[1\], \[2\] for more examples in this setting.
>
> \[1\] Performative Prediction. Perdomo et al. ICML2020.
> \[2\] Anticipating Performativity by Predicting from Predictions. Mendler-Dünner et al. NeurIPS 2022\.

---

> > ### Comment · Reviewer_Dy2p · 2024-11-26
> >
> > I appreciate the authors' responses to my questions.
> >
> > Unfortunately, the answers provided do not fully address my concerns.
> >
> > I still believe that the experiments are insufficient. For the proposed method to have practical significance in real-world scenarios, a more complex performative label shift model must be used than what was employed in the current experiments. Can the proposed method demonstrate its effectiveness as the complexity of the performative label shift model increases?

---

> > > ### Author Response · Authors · 2024-11-28
> > >
> > > Thank you for the valuable feedback! We would like to highlight that, we do not assume any functional form for the shift mechanism. Our module PaP learns the relation from $(S_{t-1}, \Lambda^{(t)})$ pairs.
> > >
> > > In response to the request, we introduced a more complex non-linear shift mechanism and re-evaluated the baselines for CIFAR100. Specifically, we used a two-layer random neural network with ReLU activation, which takes the sufficient statistic S as input and generates class probabilities for the next round. This setup increases the complexity of the shift function.
> > >
> > > As shown in Figure 8, our method successfully adapts to these more complex shifts, demonstrating PaP’s capability to predict and adjust to the outputs of non-linear mechanisms.

---

### Official Review · Reviewer_ZdNN · 2024-11-02

**Soundness:** 3
**Presentation:** 2
**Contribution:** 2
**Rating:** 3
**Confidence:** 4

**Summary:**

This paper proposed PaP to adjust pretrained backbones for performativity in a modular way, enabling the reuse of existing deep learning assets.
The  key idea is to train a shallow adapter module to perform a Bayes-optimal label shift correction to the backbone's logits given a sufficient statistic of the model to be deployed.

**Strengths:**

The paper idea is simple and reasonable. The overall framework is efficient and easy to deploy.
Compared to without adapdation, the model significantly improved the performance.

**Weaknesses:**

1. Some of the definitions in the paper is clear, which increased the difficulty to understand the paper.
2 The benchmark dataset setting is not clearly stated.

**Questions:**

1. What is the clear definition of distribution shift? There is no clear definitions for this term throughout the paper. I think in this paper you only study the label shift. You should clearly define it instead of mentioning distribution shift everywhere.
2. What is the definition of label marginals, how do we calculate it? This is also not defined but used everywhere.
3. What is the memory M in the algorithm 1? I still did not get why we need M. I did not see any definitions of it in the paper. Also, considering it is a memory buffer, I did not get the point how did you get the gradient to update T.
4.  In Table 1, PaP estimated performance is reported here. How did you calculate it? I still did not see any related descriptions in the paper. I would guess it is sufficient statistic S in your paper, but that is also never clearly defined for the related calculations.

---

> ### Author Response · Authors · 2024-11-21
> **Response to Reviewer ZdNN**
>
> >**W1.1 “**Some of the definitions in the paper is clear**”.**
>
> We have addressed your specific questions on definitions individually below. Please kindly let us know if there are any further doubts and we would be happy to clarify.
>
> >**W1.2: “The benchmark dataset setting is not clearly stated.”**
>
> In Appendix (line 882), you can see all the Dataset Details. In section 4.1 (Performative Label Shift), we defined our sufficient statistics (Equation 5\) and how we calculated the exact distribution shift given the sufficient statistic (Equation 6). In line 362, you can see the paragraph “Evaluation metric” explaining how the model’s performance on round t affects the sufficient statistic $S_t$ which leads to $P_{t+1}$. We repeatedly evaluated the baselines mentioned on lines 310-323 under the very same setting over 200 rounds.
>
> >**Q1: “**What is the clear definition of distribution shift?**”**
>
> ”Distribution shift” refers to the variation between the training and the testing distributions in general. A performative distribution shift can be a distribution shift of any kind, thus we use the general term distribution shift. While the conceptual idea of anticipating performativity based on recording a sufficient statistic applies to any distribution shift, we are very explicit that our primary focus is **label shift.** See background line 128, **where we say:**
>
> >“We primarily focus on label shift in this work. Label shift refers to the shift of the marginal distribution P(Y), while the class conditionals P(X|Y) remain fixed (Manski & Lerman, 1977; Storkey et al., 2008; Zhang et al., 2013; Lipton et al., 2018).”
>
> >**Q2: “**What is the definition of label marginals, how do we calculate it?**”**
>
> Label marginals refer to the marginal distribution of the labels P(Y) and marginalizing out the data X. It is computed empirically by counting the occurrence of each class in a given dataset.
>
> >**Q3: “**What is the memory M in the algorithm 1? I still did not get why we need M**”, “**I did not get the point how did you get the gradient to update T**”**
>
> We kindly point the reviewer to line 269:
>
> >"… we collect the statistic $S_{t−1}$ of the deployed model $f_{t−1}$, together with the induced label marginals $Λ_t$ over $P_t$ and store it in a memory buffer to learn the predictor T in a supervised manner."
>
> Having a memory buffer is a common technique for online learning algorithms and is also widely used in reinforcement learning. By storing the previously seen samples, we can create an auxiliary dataset and treat it as a classic supervised learning setting rather than online learning. Therefore, the memory buffer is our dataset and we apply simple mini-batch updates iterating the dataset.
>
> >**Q4: “In Table 1, PaP estimated performance is reported here. How did you calculate it?”**
>
> A detailed explanation of the results in Table 1 is given in the paragraph “Anticipating performativity.” (line 468). Briefly, we created balanced holdout sets on CIFAR100 (for Table 1\) and on ImageNet100 (for Table 5), and trained a pool of models where we mainly change the number of epochs and random initializations. Please also refer to the common response.
>
> To estimate the post deployment performance, we use PaP to estimate the label probabilities on the next round. Then, we sample instances from the current distribution proportionally to the predicted label probabilities, in order to simulate the next shift. Using these sampled instances, we can estimate the future performances of the models without observing the shift. We will clarify this in the draft.

---

### Official Review · Reviewer_PHw4 · 2024-11-03

**Soundness:** 3
**Presentation:** 2
**Contribution:** 3
**Rating:** 6
**Confidence:** 3

**Summary:**

This paper introduces the use of a novel adapter module with pretrained vision/language backbones to address the issues with performative label shift. These label shifts are usually a result of deployed deep learning models. The authors validated their proposed approach with simulated label shifts on vision and language tasks with adversarial sampling setting etc..

**Strengths:**

This paper addressed an important problem of performativity for deployed deep learning models given post deployment distribution shifts, and proposed a plug-and-play adapter module to be used with pre-trained backbones. The authors have formulated this interesting problem and conducted various experiments demonstrating the effectiveness of their method.

**Weaknesses:**

1. It's a bit hard to understand the proposed method given the writing, which needs to be improved particularly at places critical for elaborating the core module and results. For example, reading through section 1.1 and section 3, it's unclear what "sufficient statistic" refers to, which is an important inputs to the proposed adapter module. It's not until in later sections the authors gave examples for statistic such as class accuracies. Another example, reading through p. 7 lines 324-328, I still have trouble understanding what to take from Table 1 and how to interpret the ranking.

2. While the author claimed that none of the prior work addressing distribution shifts is designed to address them proactively. The proposed method also relies on data with label shifts to train the adapter. This method similarly requires distribution shifted data at train time. The author should clarify what is the key advantage of the adapter since it does not work proactively either.

3. It will help understand the effectiveness of the method if more experiment details are given. E.g., how does the parameter "round T" affect the final adapter from the training stage? Given that in Algorithm 1 the output is one single adapter module at time T, is this adapter used to test throughout the 100 rounds shown in Figure 2?

4. Are there some conjectures / explanations for the drastic drop in Oracle finetuning at Round 1 from Figure 2 and 3? It's unclear if this is an issue with the particular label shift or finetuning stage setup.

**Questions:**

Please see the weaknesses above.

---

> ### Author Response · Authors · 2024-11-21
> **Response to Reviewer PHw4**
>
> Thank you for your dedicated time to reviewing our manuscript. We address your comments individually below and indicate our edits thanks to your feedback. We hope that these have increased the readability of our paper and please let us know if any confusion remains. Thank you\!
>
> > **Q1.1: “it's unclear what "sufficient statistic" refers to.”**
>
> Thank you for the suggestion. We updated the document by providing an example statistic earlier.
>
> > **Q1.2 “ “have trouble understanding what to take from Table 1 and how to interpret the ranking.”**
>
> We hope the general response was able to clarify this. We updated the caption so that it is more self-contained.
>
> In summary, the table shows that the performance of models pre-deployment phase can be misleading for assessing the performance post-deployment. Depending on the self-induced shift, the ranking before the shift and after the shift can change, which creates the need of accounting for the shift. Here, we show that PaP’s estimates are aligned with the model performances after the shift and gives a good proxy on which model will perform the best.
>
> > **Q2: “**what is the key advantage of the adapter since it does not work proactively either.**”**
>
> There might be a misunderstanding here. PaP estimates the label probabilities on the next round without observing the next round samples. Thus it can anticipate what update it is going to do to the backbone’s predictions before moving on to the next round. The results in Table 1 demonstrate this ability to foresee shifts. Model 1-Model 3 are models that have not been deployed during training and still PaP can reasonably predict their post-deployment performance.
>
> > **Q3.1 and Q3.2: how does the parameter "round T" affect the final adapter from the training stage? Given that in Algorithm 1 the output is one single adapter module at time T, is this adapter used to test throughout the 100 rounds shown in Figure 2?**
>
> On Figures 1 and 2, we initialize PaP at round 1, and train it in an online manner while performing the model updates. The figures show the online performance, and we see it gets stable very fast – after the sudden drop at round 10, it quickly recovers. Therefore, the effect of T on the adapter quality is visible on the trajectory. We choose this setting purposefully to showcase how our adapter accommodates the distribution shift with only a few rounds.
>
> > **Q4: “ some conjectures / explanations for the drastic drop in Oracle finetuning at Round 1 from Figure 2 and 3”.**
>
> Figure 7 on appendix shows an ablation for Oracle-finetuning baseline with the following settings:
>
> - 5 epochs for finetuning
> - 40 epochs for finetuning
> - Training only the last linear layer
>
> However, these models are very sensitive to the distribution shifts too, mainly due to the fact that the finetuned datasets (self-induced distributions) are imbalanced.
>
> As an extreme case, If we increase the severity of the performative shift, the distribution can even oscillate between sets of classes which are disjoint, resulting in very poor performance as the model will not see the corresponding classes during its finetuning stage.

---

### Official Review · Reviewer_g3fT · 2024-11-05

**Soundness:** 3
**Presentation:** 3
**Contribution:** 3
**Rating:** 6
**Confidence:** 2

**Summary:**

This paper addresses the challenge of performative label shift, where traditional methods assume access to performativity-augmented datasets, but training from scratch for each shift can be costly. The proposed method introduces an adaptation module that predicts label marginals and applies a Bayes-optimal correction to model logits, effectively reducing performance degradation due to distributional shifts. This approach is the first to adapt deep learning models for performative shifts, offering the potential for zero-shot transfer and anticipating model brittleness across shifts. Experimental results support the method’s effectiveness in mitigating performance loss with limited performativity-augmented data.

**Strengths:**

Originality: The paper introduces two novel contributions: a practical baseline for handling performative label shift and a module to predict model performativity. These aspects offer a fresh approach in this field.

Quality: The paper is well-executed, with extensive experiments across several datasets. Baselines are clearly presented, and the rationale behind the choices is well-explained, making the contribution and results robust and grounded.

Significance: Although I am not familiar with this field, the framework appears practical, easily reproducible, and offers potential for further development. While I cannot fully assess the technical impact, the framework shows promising potential for future improvements.

**Weaknesses:**

I must admit I'm having difficulty understanding certain aspects of this article. Firstly, regarding the evaluation metric in Lines 362-363: what exactly is the 'retraining trajectory'? Is this term defined within this article, or does it originate from previous work? Additionally, how is this definition of 'Acc' distinct from 'Acc in S'? This difference significantly impacts my interpretation of the results in Figures 2 and 3.

Similarly, in Algorithm 1, does 'Deploy' imply training on the sample, and does 'Observe sample' indicate the same? I'm finding it challenging to follow the process outlined in this algorithm diagram. Furthermore, in Section 3.2, what is the purpose of the discussions on 'Dynamic benchmarking' and 'self-selection'? I haven't found results related to these methods in the experimental section.

Some details require further clarification. First, what are the results for other datasets in Table 1? This information is essential to more comprehensively support PaP's ability to predict model performativity. Second, which pre-training datasets were used, and how might the distribution of the pre-training data impact the results

**Questions:**

Please see the weakness part.

---

> ### Author Response · Authors · 2024-11-21
> **Response to Reviewer g3fT**
>
> >**Q1: “**in Lines 362-363: what exactly is the 'retraining trajectory'?**”**
>
> The “retraining trajectory” refers to the repeated training and evaluation of a specific model over multiple rounds/steps. Since the model’s training induces new distribution shifts for the next round, which in turn influences its evaluation performance on this new distribution, the performance varies over rounds. We are interested in this entire process.
>
> We have added the text “... trajectory (i.e., the repeated training and evaluation over multiple rounds/steps)” to line 363 to clarify this. Please let us know what you think.
>
> >**Q2: “**how is this definition of 'Acc' distinct from 'Acc in S'??**”**
>
> Note that we can only prepare the model using the current distribution $P_t$, but we always report the performance on $P_{t+1}$ since, when we deploy the model, the distribution changes.
>
> Therefore, Acc denotes the accuracy of the model **on the deployment distribution, signaling the start of the next round**. On the other hand, class-wise accuracies in S refer to the model's performance **in the current round.** Reported accuracies on the Figure 2 and 3 always demonstrate the model’s performance on the shifted distribution.
>
> >**Q3: “**in Algorithm 1, does 'Deploy' imply training on the sample, and does 'Observe sample' indicate the same?**”**
>
> The term deploy means releasing the model to the environment, and evaluating it on whatever distribution arises. Observe $P_t$ refers to the part where one can prepare the model for the future distribution $P_{t+1}$ using current data obtained from the distribution $P_t$.
>
> >**Q4: “**in Section 3.2, what is the purpose of the discussions on 'Dynamic benchmarking' and 'self-selection'?**”**
>
> These are two real-world examples where performative shifts arise naturally, that we wanted to discuss to showcase where our method could be applied.
>
> * **Dynamic benchmarks** are an example of a model-induced distribution shift. On these benchmarks, the model is put in a feedback loop over different distributions and iterated finetuning is applied to obtain a better performing model. Here, PaP could be used to mitigate the model-induced shift in a very lightweight way, which has interesting implications for this practice.
> * **Self-selection** is presented similarly to support an application where PaP is relevant. Model’s poor performance for subgroups can result in disengagement of users. Instead, PaP can be used very naturally to balance underrepresented classes/groups on the distribution by its reweighting strategy.
>
> >**Q5: “**First, what are the results for other datasets in Table 1?**”**
>
> We have included experiments on ImageNet100 in the Appendix (line 1047). The results are consistent with the reported results and confirm the benefit of PaP for model selection.
>
> >**Q6: “**which pre-training datasets were used, and how might the distribution of the pre-training data impact the results**?”**
>
> We created balanced holdout sets on CIFAR100 (for Table 1\) and on ImageNet100 (for Table 5), and trained a pool of models where we mainly change the number of epochs and random initializations.
>
> The distribution of the pretraining data would affect the model performance more drastically if it is now balanced. In our experimental setting, we sample from classes inversely proportionally to the model’s performance. Imbalanced datasets can lead to a poorer performance on particular classes, resulting in a more severe shift. Therefore, the gap between the pre deployment performance and post deployment performance increases.

---

> > ### Comment · Reviewer_g3fT · 2024-11-26
> >
> > Thanks for the response. For Q1, you mention that "We are interested in this entire process." I wonder if you can be more specific about what kind of trajectory or curve I should be interested in. Or, more straightforwardly, what should we expect from analyzing this process? Can you elaborate more?  For Q3, I still can not understand what "observe $P_t$ refers to the part where one can prepare the model for the future distribution $P_{t+1}$ using current data obtained from the distribution $P_t$ ' means. So, for Q4, are these two more like future works? Are there any results in these two real-world scenarios? If not, should be put into limitation and future works instead of methods.

---

> > > ### Author Response · Authors · 2024-11-28
> > >
> > > Thank you for your valuable feedback! We address your questions as below. Please kindly let us know if you have any further questions and we would appreciate the opportunity to respond properly.
> > >
> > >
> > > >**For Q1, you mention that "We are interested in this entire process." I wonder if you can be more specific about what kind of trajectory or curve I should be interested in. Or, more straightforwardly, what should we expect from analyzing this process? Can you elaborate more**
> > >
> > > Thank you for your thoughtful question.
> > >
> > > We analyze the model accuracy trajectory over multiple training rounds, as depicted in Figure 2 of our manuscript. Specifically, we focus on two key aspects:
> > > 1. **Average Accuracy:** This measures the mean performance of the model across all rounds.
> > > 2. **Stability of Accuracy:** This assesses how consistent the model's performance remains over successive rounds.
> > >
> > > A high and flat accuracy curve signifies that the model not only maintains strong performance but is also robust to performative distribution shifts, such as Oracle distribution and PaP in Figure 2.
> > >
> > > **Expectations from Analyzing the Trajectory:**
> > > We calculate these accuracies by evaluating the current model $f_t$ on the shifted distribution $P_{t+1}$. The shift from $P_t$ to $P_{t+1}$ is induced by the model $f_t$, which is trained on $P_t$ (as outlined in Equation (6)). This shift is adversarial in nature (refer to line 353), meaning it is designed to challenge the model's adaptability within the setup shown in Figure 2.
> > >
> > > By examining this accuracy trajectory, we aim to understand how well the model adapts to changing data distributions over time. Specifically:
> > > - Robust Models: Should exhibit stable and consistently high accuracy, indicating resilience to shifts.
> > > - Non-Robust Models: May show fluctuating or declining accuracy, especially under high-shift regimes.
> > >
> > > **Comparison:**
> > > The baseline Oracle Fine-Tuning approach exhibits sharp performance declines and struggles to recover in high-shift regimes (as seen in the top rows of Figure 2). This decline occurs because Oracle Fine-Tuning solely relies on fine-tuning on $P_t$ without any proactive adaptation to anticipate future distribution shifts.
> > >
> > > In contrast, PaP effectively estimates the future distribution $P_{t+1}$ to adapt accordingly. This allows PaP to nearly match the performance of the Oracle which directly utilizes the ground truth $P_{t+1}$. The ability to proactively forecast and adapt to future shifts is essential in high-shift regimes, where traditional methods like Oracle Fine-Tuning fall short.
> > >
> > >
> > >
> > > >**For Q3, I still can not understand what "observe $P_t$ refers to the part where one can prepare the model for the future distribution $P_{t+1}$ using current data obtained from the distribution $P_t$ ' means.**
> > >
> > > Thanks for pointing us to this specific question. In the context of Algorithm 1, "Observe samples from $P_{t}$" (Algorithm line 4) refers to obtaining samples from the current distribution $P_{t}$. We extract statistics $\Lambda _{t}$ from these samples (Algorithm line 6), which is further employed for training our adapter for this round $t$ (Algorithm line 8).
> > >
> > > For comparison, the baseline Oracle Fine-tuning, also utilizes these samples (image label pairs sampled from $P_{t}$) to directly fine-tune the backbone.
> > >
> > > Our empirical results show that our approach—predicting the next label distribution to update the backbone’s predictions—proves more effective and computationally efficient than retraining the backbone itself.
> > >
> > >
> > > >**For Q4, are these two more like future works? Are there any results in these two real-world scenarios? If not, should be put into limitation and future works instead of methods.**
> > >
> > > Thank you for the constructive feedback! These two applications can be two practical instantiations of our framework, to which our method can directly be applied, so they are not future work. For example, the adversarial label shift setup in our evaluation can be one form of the dynamic benchmark [1].
> > >
> > >
> > > [1] Yixin Nie, Adina Williams, Emily Dinan, Mohit Bansal, Jason Weston, and Douwe Kiela. Adversarial NLI: A new benchmark for natural language understanding. In Proceedings of the 58th Annual Meeting of the Association for Computational Linguistics, pp. 4885–4901, 2020.

---

### Author Response · Authors · 2024-11-21
**General response to all reviewers and the AC**

We would like to thank all reviewers for their feedback and suggestions, especially pointing out concepts that could benefit from further clarifications.

Below we briefly clarify some terminology before addressing the questions and concerns of each reviewer individually.

**Model evaluation:** During the training phase we repeatedly evaluate models under the self-induced distributions. Thus, the sequence of distribution observed during training is determined by how models are updated across rounds. We refer to this sequence as “retraining trajectory”. Naturally, this trajectory will be different depending on the strategy used to update models. While fine-tuning adjusts the model based on data from the current distribution, the goal of our algorithm is to prepare the model for the distribution shift it will see in the next round.

We refer to **pre-deployment** performance as the performance evaluated offline on the current distribution induced by the model deployed in the previous round and **post-deployment\*\* as the performance of the model on the distribution it induces. **Deployment** is the action that causes the shift. Thus, we distinguish between model updates that are made offline and deployments that have a causal effect on the distribution. We hope this clarifies the terminology.

**Comments on Table 1:** Here, we evaluate the performance of the end product of the training phase. We want to see how useful the learned adaptation model is for anticipating unseen shifts and foresee the performance of future deployments. To this end, we select three candidate models to be deployed. We compare the performance of these models as it appears to be on the current data (pre-deployment) and the performance as it actually is after the shift (post deployment). In this case, we see that the relative ranking of models is changed by the shift, and our module (PaP) is able to foresee this and correctly select the best model **pre** deployment. Instead, performing model selection based on the in-distribution performance might lead us to pick the least performing model.

The ability to foresee the performance after the shift is the main goal of the adapter module. Our results show that it is possible to learn the relationship between the statistic and the induced shift from data collected along a natural retraining trajectory to then foresee the consequences of future models (not seen during training\!). We are the first to demonstrate the possibility to anticipate performativity with neural networks. As requested by the reviewer g3fT, we additionally confirm these results on ImageNet100, in Table 5.

**Note:** All modifications in the manuscript are highlighted with a **blue** color. We are happy to address further questions or concerns during the rebuttal period.

---

### Author Response · Authors · 2024-12-02

As the discussion period deadline approaches, we kindly ask reviewers to let us know if any further clarifications are needed. We are happy to provide additional details or address any concerns.

---

### Meta-Review · Area_Chair_Rohx · 2024-12-20

**Metareview:**

This work addresses the performative label shift without re-training the model. The proposed method introduces an adaptation module that predicts label marginals and applies a Bayes-optimal correction to model logits. The reviews are mix, and the authors failed addressing the concern over insufficient experiments, since the assumed performative label shift model is too simple. Thus, I recommended a rejection.

**Additional Comments On Reviewer Discussion:**

The reviewers did not change the score after rebuttal.

---

### Decision · Program_Chairs · 2025-01-22

Reject